# Long Non-Coding RNAs: New Insights in Neurodegenerative Diseases

**DOI:** 10.3390/ijms25042268

**Published:** 2024-02-14

**Authors:** Adithya K. Anilkumar, Puneet Vij, Samantha Lopez, Sophia M. Leslie, Kyle Doxtater, Mohammad Moshahid Khan, Murali M. Yallapu, Subhash C. Chauhan, Gladys E. Maestre, Manish K. Tripathi

**Affiliations:** 1Medicine and Oncology, ISU, University of Texas Rio Grande Valley, McAllen, TX 78504, USA; 2South Texas Center of Excellence in Cancer Research, School of Medicine, University of Texas Rio Grande Valley, McAllen, TX 78504, USA; 3Department of Pharmaceutical Sciences, St. John’s University, Queens, NY 11439, USA; 4Department of Neurology, College of Medicine, University of Tennessee Health Science Center, Memphis, TN 38163, USA; 5Department of Neurosciences, University of Texas Rio Grande Valley School of Medicine, Brownsville, TX 78550, USA; 6South Texas Alzheimer’s Disease Research Center, School of Medicine, University of Texas Rio Grande Valley, Harlingen, TX 78550, USA

**Keywords:** long non-coding (Lnc) RNAs, neurodegenerative diseases (NDDs), Alzheimer’s disease (AD), Parkinson’s disease (PD), amyotrophic lateral sclerosis (ALS), biomarkers

## Abstract

Neurodegenerative diseases (NDDs), including Alzheimer’s disease (AD), Parkinson’s disease (PD), and amyotrophic lateral sclerosis (ALS), are gradually becoming a burden to society. The adverse effects and mortality/morbidity rates associated with these NDDs are a cause of many healthcare concerns. The pathologic alterations of NDDs are related to mitochondrial dysfunction, oxidative stress, and inflammation, which further stimulate the progression of NDDs. Recently, long non-coding RNAs (lncRNAs) have attracted ample attention as critical mediators in the pathology of NDDs. However, there is a significant gap in understanding the biological function, molecular mechanisms, and potential importance of lncRNAs in NDDs. This review documents the current research on lncRNAs and their implications in NDDs. We further summarize the potential implication of lncRNAs to serve as novel therapeutic targets and biomarkers for patients with NDDs.

## 1. Introduction

Neurodegeneration is the progressive deterioration of the body’s central communication tools and, thus, the very essence of sentient life. There are limited diagnostics and treatment regimens that exist to combat the complex mechanisms at the core of neurodegenerative diseases (NDDs) [1,2]. Although disease pathogenesis and progression mechanisms have been relatively well studied, causative agents remain elusive for most NDDs [3]. Moreover, postmortem examinations of brain tissue are the gold standard method for accurate confirmation of many NDDs [2,4]. A common tie between the vast majority of NDDs are age-related changes at the cellular level [1]. Based on a United Nations 2020 report on the aging global population, there is a reasonable cause for concern for the prevalence of NDDs, with the number of persons aged >65 years having been projected to double by 2050 [5]. The more widely studied, albeit debilitating, NDDs include Alzheimer’s disease, Parkinson’s disease, and amyotrophic lateral sclerosis (ALS). These three are unique in their pathophysiology but notorious for gradually worsening of symptoms. There are various known biomarkers for the diagnosis of these NDDs. Still, the poor prognostics and difficulty of assessments for Parkinson’s, Alzheimer’s, and ALS diagnoses call for the development of biomarkers that can aid in early detection and treatment to prevent disease progression [2]. Therefore, it is important to improve our understanding of molecular mechanisms underlying neurodegeneration and identify new therapeutic targets for the prevention and treatment of NDDs. Figure 1 shows the general mechanisms of NDDs.

Progressive loss or atrophy of functional neurons or disruptions to neuronal connections is often associated with aggregates of misfolded proteins, resulting in structural and functional impairments [1,3]. One study listed hyperphosphorylated-τ-immunoreactivity (HPτ-IR), β-amyloid immunoreactivity (Aβ-IR), α-synuclein immunoreactivity (αS-IR), and transactive response DNA-binding protein 43 (TDP43) as common concomitant misfolded protein aggregates in a senior cohort of cognitively unimpaired subjects. Thus, it is imperative to frequently assess regions of the brain that have high distributions of these proteins for lesions [6]. Notably, proteins like HPτ-IR, whose changes in pathology or expression have been documented in the neurodegeneration of cerebral white matter, make valuable biomarkers for more accurate diagnoses [6,7,8]. The domineering features of NDDs include Lewy bodies, neurofibrillary tangles, and amyloid plaques, all showing a gradual increase with age [9]. It should be noted, however, that aging is not a linear pathway on its own but rather the aggregation of individual bodily processes, including environmental influences, epigenetic changes, or inherent dysregulation in gene expression due to DNA damage [9]. Increasing amounts of evidence focus on RNAs as contributing factors in NDDs. Also, extensive research into non-coding RNA functions has expanded our understanding of diverse pathological processes over the past few decades. One of the vital ncRNAs is long non-coding RNAs (lncRNAs) [10].

LncRNAs are novel genetic biomarkers that can be used as exclusionary tools specific to NDDs. These historical biomarkers have been there for years, so a change in the approach is necessary to better diagnose and treat these NDDs. The concept of lncRNAs was not well known earlier but, in recent years, it has emerged as an indispensable player in the diagnosis, development, and therapeutics of NDDs [11].

**Figure 1 ijms-25-02268-f001:**
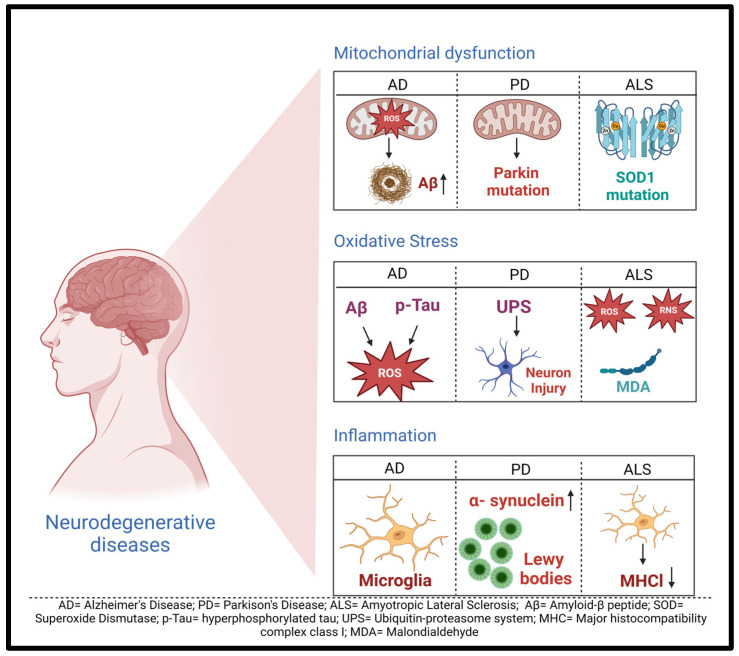
General mechanisms of neurodegenerative diseases. The most important mechanisms are mitochondrial dysfunction, oxidative stress, and inflammation. (1) Due to mitochondrial dysfunction, Aβ deposition occurs through reactive oxygen species (ROS) production in AD. It can also lead to Parkin mutations in PD, whereas the cause of ALS is SOD1 mutation. (2) Due to oxidative stress/imbalance, the production of Aβ and p-Tau occurs in AD. It causes UPS dysfunction, which augments dopaminergic neuronal injury in PD, and there is an overproduction of reactive oxygen and nitrogen species along with MDA in ALS. (3) Microglia regulate inflammation during the innate response of the CNS in AD. The accumulation and aggregation of α-synuclein in Lewy bodies are involved in PD development. The microgliosis process can result in a decline in the level of MHCl, which contributes to ALS progression [12]. Created with BioRender.com.

The discovery of non-coding RNAs has called into question the integrity of the central dogma, which delineates RNA as a mediatory vehicle from DNA to protein [13]. Despite their lack of protein-coding elements, overwhelming evidence has identified the roles of ncRNA in regulating cellular processes as well as implications in disease development and progression [10,14,15,16]. This review aims to explore how the specific expression patterns at the tissue and cellular levels of lncRNA make them ideal for therapeutics and for diagnostic or prognostic tools to probe for evasive NDDs.

## 2. Long Non-Coding RNAs: An Overview

Long non-coding RNAs, classified as non-coding RNA composed of more than 200 nucleotides, have been highlighted as important biomolecules in the cellular mechanisms responsible for normal development to disease progression. LncRNAs, being just one of the types of non-coding RNA, can be further divided into multiple categories by their genomic position. These categories include intergenic lncRNAs, antisense lncRNAs, intronic lncRNAs, and bidirectional lncRNAs [17]. Along with varying genomic position, the localization of lncRNAs can also differ with studies showing nuclear and cytoplasmic localization [18,19]. The primary mechanism that drives nuclear localization is unknown, but it can potentially be attributed to the recruitment of nuclear retention factors facilitated by sequence motifs [20,21]. When these sequences were removed, lncRNA exportation was favorable, and cytoplasmic localization was possible. Compared to mRNA, the structure and biogenesis of lncRNA are similar due to the addition of poly-A tails, 5′ 7-methylguanoasine capping, and transcription by RNA Polymerase II [22]. Unlike mRNA, however, lncRNAs are relatively less abundant, lack the open reading frame typically found on mRNA, and, most importantly, lack a protein-coding ability [22]. Previously, this last feature of lncRNA, along with other non-coding RNA, led to their classification as “junk” DNA with no discernible function. This has been refuted by recent research showing the roles that lncRNAs plays in various biological processes, such as the progression of cancer, neurodegenerative diseases, and normal development [23,24,25]. It has been shown that lncRNAs play a crucial role in the regulation of diseases course and normal cellular processes through gene regulation. Figure 2 shows the schematic representation of lncRNA function.

Additionally, lncRNA localization correlates highly to function, with nuclear lncRNA functioning primarily as chromatin regulators, while cytoplasmic lncRNAs are more focused on post-translational regulation and the stability of mRNA [21]. LncRNAs have the potential to modulate normal and abnormal development through their mechanisms as decoys, scaffolds, and guides, in addition to direct interaction with DNA and protein molecules [26].

## 3. LncRNA Mechanisms

LncRNAs play a vital role in disease progression and development that is highly dependent on their well-known functions, including altering gene expression by modulating chromatin structure and regulating transcription and post-transcriptional modifications [27]. These processes can be achieved by lncRNAs acting as guides, decoys, and scaffolds [26,28]. LncRNAs can function as guides and recruit chromatin-remodeling complexes to a specific locus in cis or trans sites [29]. Through recruitment, lncRNAs can target specific sections to alter the structure of chromatin and, therefore, alter the gene expression of that specific gene. LncRNAs can also directly interact with DNA to form hybrid structures such as R-loops or triple helices (triplexes). Triple helices have the potential to mediate gene activation or repression by recruiting remodeling complexes [30]. R-loops can be recognized by a variety of proteins to also promote or suppress gene expression [31]. Furthermore, lncRNAs can interact with proteins and function as scaffolds. As a scaffold, they can aid in the binding of proteins to assemble a complex commonly known as ribonucleoprotein complexes [32]. These complexes have roles in mRNA splicing, translation, and stability, allowing lncRNAs to affect these processes indirectly [18]. LncRNAs can also function as decoys and lower the availability of regulation factors, which can negatively impact gene expression [33]. Furthermore, lncRNAs can regulate pathways and expression through miRNA sponging. This mechanism, enables lncRNAs to act as an miRNA target and bind to miRNA. The result of miRNA sponging is a reduction in the miRNA function, leading to the alteration of signaling pathways that can affect further gene expression [34].

## 4. LncRNAs in Neurodegenerative Diseases

### 4.1. Alzheimer’s Disease

Alzheimer’s disease (AD) is a progressive and irreversible neurodegenerative disorder that is the most common form of dementia. This disease is named after Dr. Alois Alzheimer, who noticed that there had been significant changes in the brain of a woman who had unusual symptoms such as memory loss, language problems, and unpredictable behavior. These significant changes are now known as two hallmark characteristics of AD. The first is amyloid plaques, which are produced from amyloid β (Aβ) aggregation, and the second is neurofibrillary tangles (NFT), which are produced from an accumulation of pathological tau [35]. Late-onset AD (LOAD) usually presents as an age-related disease in individuals 65 and over. There is a small portion of individuals with AD who have early-onset Alzheimer’s disease (EOAD), which occurs in people younger than 65 with a genetic predisposition. In the hundred or so years since the discovery of the disease, there have been many studies looking to elucidate the mechanisms of the disease. Although treatments may help relieve some of the symptoms and/or progression associated with NDDs, there are currently no known cures. Therefore, there is a critical need to expand our understanding of what causes neurodegeneration and to develop new approaches for the prevention and treatment of AD [35].

In recent studies, lncRNAs have been shown to be involved in regulatory mechanisms such as transcriptional, posttranscriptional, and translational regulation, as well as a variety of biological functions such as development, differentiation, and metabolism [36]. LncRNAs have also been shown to be involved in neurological diseases such as epilepsy, neurodegenerative diseases, and genetic disorders [37,38,39]. LncRNAs have also been shown to play a role in the pathogenesis of AD, yet the exact mechanism is still unknown. In a microarray analysis study of lncRNAs expressed in the hippocampal region of AD, 315 lncRNAs were found to be dysregulated [40]. Many protein-coding mRNAs have antisense transcript “partners,” which are commonly noncoding RNAs. An example would be the antiscript BACE1-AS, which has been shown to regulate *BACE1* mRNA and its protein expression. BACE1 (beta-site amyloid precursor protein cleaving enzyme1) is required for the formation of all monomeric Aβ (1-42) and is also thought to be the cause of toxicity in AD patients [41]. This study shows that lncRNAs are responsible for the increase in Aβ 1-42 in AD, showing that lncRNAs influence the pathogenesis of AD [42].

#### 4.1.1. Competitive Endogenous RNA (ceRNA) Theory

Several studies have reported that lncRNA competes with miRNA target genes by sharing common binding sites [38]. Based on the ceRNA theory, Wang and collaborators created a global triple network where lncRNAs and mRNAs form a triplicate that shares the same miRNA. Based on this network, an AD NFT-associated lncRNA-mRNA network (NFTLMN) was created, mapped, and analyzed, providing three lncRNAs highly related to AD NFT [38]. Gene ontology (GO) function and KEGG pathway enrichment analyses were performed on AP000265.1, KB-1460A1.5, and RP11-145M9.4, showing GO terms for formation and development of the neural tube, neural crest cells, and epithelial tube morphogenesis. Phosphorylation terms were also found during the analysis [38]. A different study found 40 pairs of lncRNAs that shared more than one disordered miRNA, 9 of them correlated with other neurodegenerative disorders, and 5 lncRNAs that could be potential biomarkers for [43]. These machine learning studies have found several lncRNAs that would be suitable for further research as possible biomarkers since they have been identified as showing a correlation with AD genes, but their exact function is unknown.

#### 4.1.2. LncRNA Involvement

Many lncRNAs have been identified as having a role in AD. Although many of these newly identified lncRNAs have been found through bioinformatics, many of these lncRNA functions remain obscure. The lncRNAs related to AD that have been studied have been discovered to be involved in synaptic and neuron exhaustion, neurotrophin depletion, inflammation, mitochondrial impairment, oxidative stress, and DNA damage [44]. lncRNAs, such as NDM29, BC200, 51A, and BACE1-AS, are differentially expressed in AD and correlated with AD progression [45]. Neuroblastoma differentiation marker 29 (NDM29), a lncRNA that promotes the cleavage of BACE and γ-secretase, plays a critical role in AD pathogenesis by inducing an inflammatory response. Studies show NDM29 can induce APP synthesis, increasing Aβ and Aβ-42/Aβ-40 ratio [44,45]. The BACE1 antisense transcript (BACE1-AS) regulates BACE1 mRNA and protein expression when exposed to Aβ-42 [44]. BACE1-AS is significantly upregulated in the cerebellum, hippocampus, entorhinal cortex, and superior frontal gyrus of the AD brain. There is a synergistic mechanism in how BACE1-AS regulates BACE1, which can promote target mRNA or inhibit miRNA [46]. MALAT1 is a highly abundant and evolutionarily conserved lncRNA and regulates a subset of genes involved in synaptic plasticity. Recent studies have reported reduced lncRNA MALAT1 levels in the central spine fluid and brains AD patients compared with a control group [47,48]. Several preclinical studies have provided evidence and support for the potential roles of lncRNA MALAT1 in AD pathogenesis. For instance, studies in experimental AD models indicated enhanced neurite outgrowth, reduced proinflammatory cytokines, decreased neuronal apoptosis, and increased presynaptic bouton density on dendrites with lncRNA MALAT1 overexpression, and vice versa with lncRNA MALAT1 knockdown [49]. Similarly, a recent study suggested the beneficial effect of lncRNA MALAT1 against Aβ1-42-induced toxicity [47]. lncRNA-51A overlaps with SORL1 antisense and could affect amyloid beta formation, which is known to be upregulated in AD [38]. Sortilin-related receptor 1 (SORL1) is one of the genes involved in the processing of amyloid-β protein precursor (APP) and is thought to be a genetic factor of AD. LncRNA-51A downregulates SORL1, which results in abnormal APP processing [50]. BC200 exhibits abnormal subcellular localization and expression levels in specific brain regions in AD patients [37]. BC200 is a protomer-associated RNA that works on the translation level by increasing synapse loss [44]. The loss of synapses is one of the pathological features of AD and is the cause of memory loss in AD patients. In normal aging, BC200 was found to be downregulated, but in AD brains it was found to be significantly upregulated in brain areas related to AD. Furthermore, relative levels of BC200 RNA in AD-affected areas of the brain increased according to the severity of the disease [51]. Li et al., found BC200 could be a positive regulator of BACE1 in AD [52]. Many studies have shown that lncRNA nuclear paraspeckle assembly transport 1 (NEAT1) promotes inflammation and has a role in neurodegenerative disorders. NEAT1 is upregulated in AD and progresses via the miR-124–BACE1 axis showing it can be manipulated to modulate BACE1 expression. This information demonstrates that NEAT1 can be a target for pharmacological therapies and a biomarker for the disease [53]. With the continuing bioinformatic research, several lncRNAs have been identified with tentative functions and correlations. A study by Nana Ma and collaborators identified 487 significantly dysregulated lncRNAs in AD model mice (APP/PS1) brains. These lncRNAs were found to be involved in synaptic plasticity and memory (Akap5), and regulation of amyloid-β induced neuroinflammation (Klf4) [54]. Transcriptomic analysis identified RP3-522J7, MIR3180-2, and MIR3180-3 as lncRNAs which were most highly co-expressed with known AD-related genes [55]. Shi et al., also investigated genomic localization and found that several lncRNAs are located near important protein-coding genes (PCGs) in the human genome. For example, six differentially expressed lncRNAs were within 10 MB of the PCG S100B [55]. These non-coding RNAs have been identified by machine learning and, as such, have no identified function at this time. Figure 3 shows the lncRNAs involved in Alzheimer’s disease.

### 4.2. Parkinson’s Disease

Parkinson’s disease (PD) is a progressive neurodegenerative disease that is characterized by death or malfunction of dopaminergic neurons in the substantia nigra and dopamine depletion in the striatum resulting in the loss of motor and non-motor functions. Approximately 60,000 new PD cases are diagnosed each year, joining the 1 million Americans who currently have PD. The direct and indirect annual medical costs for PD approach $25.4 billion and $26.5 billion, respectively. Despite having gained considerable knowledge about the pathological mechanisms of PD over the last several decades, we know little about how to stop or delay the ongoing neurodegenerative processes. Symptoms typically occur when approximately 70% of the neurons are lost, and other characterizations, such as Lewy bodies composed of α-synuclein, can accumulate in the substantia nigra [56]. Currently, there is no effective cure for PD, and most medications are prescribed for symptom management. Methods of increasing striatal dopamine, such as levodopa, dopamine antagonists, and monoamine oxidase B inhibitors, are used to treat the motor implications of Parkinson’s disease [57]. Although these medications help alleviate motor symptoms, they do not aid in slowing disease progression. Thus, PD patients are in urgent need of disease-modifying therapies that can slow or stop the progression of the neurodegenerative process. The exact molecular mechanisms of PD development are not known. However, lncRNAs have been highly regarded as potential regulators for Parkinson’s disease. This potential is due to their role in biological processes and studies showing altered pathways in PD. Specific lncRNAs have been identified as altering expressions in PD patients compared to controls in samples including brain tissue, blood, and cerebrospinal fluid. Some examples of these lncRNAs include AL049437 and AK021630, which were found to be significantly upregulated and downregulated, respectively, in PD [58]. Mechanistically, lncRNAs found to be involved in PD pathogenicity aid or worsen the disease through various pathways, with most studies targeting primary characteristics in the pathophysiology of PD such as neuronal injury, inflammation, and α-synuclein accumulation.

#### LncRNA Involvement

Neuronal damage is prevalent in Parkinson’s disease, with most symptoms resulting from the loss of dopaminergic neurons. Some lncRNAs are involved in regulating injury through autophagy and the apoptosis of neuronal cells. LncRNA NEAT1 has been found to be upregulated in mice with MPTP-induced PD, which promotes the stability of PINK1 expression. As a result, it promotes the autophagy caused by MPTP to induce PD [59]. Alternatively, NEAT1 can regulate PD progression in MPP+ SK-N-SH cells by functioning as an miRNA sponge for miR-212-3p and, therefore, modulating the expression of AXIN1 protein to mediate cellular apoptosis [60]. Similarly, lncRNA BDNF-AS, when upregulated, works by downregulating miR-125b-5p to regulate autophagy and apoptosis in cells treated with MPP+ [61]. H19 is also involved in autophagy regulation, where MPP+-induced apoptosis was reduced in H19 overexpressing cells through the negative regulation of miR-585-3p [62]. Additionally, H19 overexpression in induced PD can inhibit the function of miR-301b-3p. When inhibited, it increases the expression of HRPT, usually deficient in PD, through the Wnt/B-catenin pathway, decreasing the loss of dopaminergic neurons [63]. LncRNA Xist is also involved in increasing neuronal injury through the repression of miR-199a-3p. This act allows Xist to induce the expression of transcription factor Sp1, which promotes LRRK2 and has been shown to advance PD progression [64]. Furthermore, lncRNA HOTAIR, which is found to be upregulated, furthered PD through an increase in ROS generation and neuroinflammation; therefore, inducing neuronal injury. HOTAIR potentially increases neuronal injury by regulating the autophagy protein ATG10. The protein expression is promoted through HOTAIR, acting as a sponge for miR-874-5p [65]. Other lncRNA studies have focused on the role lncRNAs play in significant inflammation in PD cases, often regulated by microglial cells, and how this can lead to the progression of neuron injury [66,67]. Patients with PD tend to have a consistent inflammatory response, which could worsen neuron injury [68]. LncRNA MALAT1 was upregulated in MPTP-treated mice, along with increased expression of pro-inflammatory cytokines through epigenetic regulation. In BV2 cells, MALAT1 epigenetically regulated Nrf2 by binding to EZH2. When the expression of Nrf2 is reduced, it increases ROS and inflammation leading to the injury of neurons [69]. LncRNA TUG1 was upregulated in PD-induced mice along with pro-inflammatory cytokines IL-6 and TNF-α [70]. LncRNA UCA1 influences oxidative stress and the expression of TNF-α, IL-6, and IL-1β. In the 6-OHDA PD mouse model, a downregulated UCA1 reduced activation of the PI3K/AKT pathway. The PI3K/AKT pathway is usually involved in a variety of cellular pathways, including neurodegenerative diseases such as PD [71]. Dysregulation of α-synuclein, commonly found in PD, has been correlated to impairment in a variety of cellular processes, such as synaptic vesicles, mitochondria function, and the autophagy-lysosomal pathway, leading to a problem in dopamine levels [72]. These factors make it a therapeutic target for studies determining the involvement of lncRNA regulation and improving PD pathogenesis. LncRNA SNHG1 overexpression reduced miR-15-5p expression and promoted α-synuclein accumulation through the SIAH1 protein. Furthermore, overexpression of lncRNA OIP5-AS1 reduced the α-synuclein accumulation by miR-126 binding and PLK2/α-synuclein autophagy [73]. In a rotenone-induced PD mouse model, lncRNA SNHG14 was discovered to be upregulated with a reversed expression of miR-133b. Upregulation of SNHG14 correlated with increased neuronal injury and an increased a-syn expression through the downregulation of miR-133b [74]. In addition, lncRNA-T199678 expression, through binding and regulation of miR-101-3p, can regulate ROS and apoptosis that was induced by α-syn [75]. Figure 4 shows the lncRNAs involved in Parkinson’s disease.

### 4.3. Amyotrophic Lateral Sclerosis

Amyotrophic lateral sclerosis, also known as Lou Gehrig’s disease, is a progressive neuromuscular degeneration that primarily affects motor neurons of the somatic nervous system. The gradual erosion of these neuromuscular connections, often starting in distal muscles, encompasses loss in much of the motor functions necessary for daily tasks [76]. Deterioration of the efferent pathways from the brain and spinal cord to effectors incites voluntary muscular atrophy. Though some variation exists in the age of onset, epidemiological studies have shown a positive correlation between age and the number of ALS cases, with the average age of diagnosis being between 55 and 65.1 [77]. Although a generally rare neurodegenerative disorder, ALS has an incidence rate of 2 in 100,000 people annually but increases for older individuals [76,77,78]. Approximately 10–15% of cases result in dual diagnoses of ALS and concomitant frontotemporal dementia (FTD), which shows a wide spectrum of overlapping symptoms and genetics [14,76]. The key differences between ALS and FTD are the targeted locations of deterioration.

Degeneration of the frontal and temporal lobes often corresponds with a range of behavioral changes, whereas ALS is defined by the deterioration of upper and lower motor neurons of the motor cortex that subsequently result in muscle paralysis and atrophy [14]. ALS is known to present itself sporadically (sALS) or genetically. Familial ALS (fALS) constitutes 5–10% of overall cases, where a single allele from a myriad of disease-causing genes will suffice for its onset due to its autosomal dominant pattern of inheritance [76,79]. Unfortunately, individuals suffering from ALS are projected to live 2–5 years after its onset [80]. Genes notorious for their involvement in the development of ALS include mutated or dysregulated ALS2, NEFH, C9orf72, SOD1, FUS, and TARDBP [2,76,81,82].

While genetic biomarkers can provide an assessment of the risk of ALS development, significant heterogeneity has also been associated with the disease, and their exact functions in ALS development and progression are unclear [76,82,83]. Additionally, many other factors are engaged in disease development and progression, including epigenetics, environment, and age-related issues such as oxidative stress [2]. Nevertheless, the average patient dies within 2–5 years [76]. Clinicians are currently restrained from providing palliative care with anti-excitatory drugs to improve patient quality of life, as there is no cure for ALS [79]. With a lack of consensus on standard and specific biomarkers, ALS cases continue to pose a grave threat. The severity of late-stage disease and ambiguity of early-stage symptoms of ALS sets grounds to demand more accurate diagnostic and prognostic biomarkers. LncRNAs have been implicated in regulatory mechanisms leading to ALS development and progression and show high tissue specificity, making them prime targets for more effective diagnostics and therapeutics [84].

#### LncRNA Involvement

Several studies have shown abnormal RNA metabolism from mutagenic RNA-binding proteins FUS, C9orf72, and TDP-43 to be an innate characteristic of ALS pathogenesis [84,85,86]. Paraspeckles are essential components with protective roles in the cellular stress response of motor neurons (MNs). A functional aspect of paraspeckles is their involvement in nucleoplasmic sequestration of RNA and proteins that directly alters target site expression [87]. The aggregation of paraspeckles in the CNS is a hallmark feature of ALS. LncRNA NEAT1 has inherent roles as a scaffold for paraspeckle formation [85,88]. Enriched lncRNA NEAT1_2 is proved to be the target of both FUS and TDP-43 through its UG-rich sequences; however, mutagenic or dysregulated RNA-binding proteins are associated with distorted or hyper-assembly of paraspeckles in early ALS pathogenesis [86,89]. In vivo studies have identified NEAT1 involvement in specific neurodegenerative pathways such as inflammation and neuronal cell death through p53 regulation; hence, the decreased brain density seen in brain scans and postmortem examinations of many NDD patients [81]. Of course, paraspeckles are primarily absent in healthy MNs because of decreased expression of NEAT1 isoforms essential to their assembly in the central nervous system (CNS), as studied with in vitro post-mitotic neurons [85,89]. Furthermore, NEAT1 expression has roles in neuron-specific pathways, and the aberrant hyperexcitability of affected MNs has been implicated in ALS and other NDDs [85]. The sequestering of genetic regulatory elements, including miRNAs by paraspeckle formation through abnormal NEAT1 expression in response to stressful external stimuli, such as proteasomal inhibition and viral infections, have also been identified in ALS [90]. These findings confer lncRNA NEAT1 in the CNS to be a viable target for ALS therapeutics.

Other discoveries of ncRNA’s influence in the pathophysiology of NDDs include antisense C9orf72 transcripts that have been connected to chromosome 9p-linked ALS. The C9orf72 antisense transcript appears to be highly conducive in fALS development, with 22.5% of fALS cases attributable to the hexanucleotide repeat expansion [91]. Experimental findings show various disease mechanisms the transcript acts through, though it has a significant role in nuclear RNA sequestration, which directly impacts transcription [92]. LncRNA CCND1, a FUS-bound transcript and cell proliferation regulator, is another ncRNA associated with ALS, particularly through the Wnt/β-catenin signaling pathway [93,94,95]. FUS acts as an inhibitor to CCND1 expression to regulate cell cycle progression; however, its dysregulation may trigger apoptosis in ALS [96]. Figure 5 shows the lncRNAs involved in ALS.

Dingsheng et al. generated an ALS-specific competitive endogenous RNA (ceRNA) network to investigate RNA transcripts with sponging effects on miRNAs to differentially regulate their expression in ALS [98]. Notably, MALAT1 was found to act as a sponge to modulate the expression of 75 genes, 7 of which have been connected to the pathogenesis of ALS by interacting with miRNA [98]. Furthermore, MALAT1, like NEAT1, is bound by TDP-43, an RNA-binding protein involved in ALS [99]. TDP-43 is noted to be a causative agent of mitochondrial dysfunction with implications in neuroinflammation, a common feature of early-stage ALS. MALAT1 is also known to regulate hnRNPA2/B1 and XIAP, both of which are closely related to apoptotic pathways that lead to neurodegeneration in ALS [98]. Experimental findings show that MALAT1 regulates the ATM gene, which is a member of the p53 apoptotic pathway in response to the DNA damage linked to the pathogenesis of ALS. Moreover, the endocytic pathway protein AAK1, which associates with SOD1 mutants and has been implicated in ALS, has also been identified as a regulatory target of MALAT1 [98]. Evidently, MALAT1 is another crucial player that can potentially have a role as a therapeutic target and a potential biomarker for ALS assessments or targeted therapies.

While a large pool of research is accessible for coding transcripts, detailed investigations of lncRNA involvement in ALS pathogenesis are in their infancy. Nevertheless, recent literature introduces differential expression of lncRNAs with impacts on transcriptional regulation pathways. ZEB1-AS1, ZBTB11-AS1, and XXbac-BPG252P9.10 are reported as novel antisense transcripts involved in ALS transcriptional regulation [41]. ZEB1-AS1 gets its significance as an antisense transcriptional regulator of Zinc Finger E-Box Binding Homeobox 1 (ZEB1), which is a highly conserved transcriptional repressor with roles in chromatin and E-Box binding; however, ZEB1-AS1 was downregulated in sALS samples in comparison to healthy control groups [98]. ZBTB11-AS1 is another differentially expressed antisense transcript coupled with the Zinc Finger and BTB Domain Counting 11 gene (ZBTB11), implying its participation in transcriptional regulation. Interestingly, cases of sALS presented with downregulated ZBTB11-AS1 [100]. XXbac-BPG252P9.10 is associated with the nuclear factor-kappa-B/REL (NF-κB) transcription factor family. The NF-κB proteins have critical roles in inflammatory and survival pathways but, in particular, IER3 is known for its regulation of anti-apoptotic genes and roles in ALS. LncRNA XXbac-BPG252P9.10 is an antisense regulator of IER3 and is significantly downregulated in sALS samples [101]. Non-coding antisense RNA expression analyses and co-expression networks involving lncRNAs and mRNAs show a multitude of potential transcripts with involvement in ALS pathogenesis that may prove to be promising therapeutic or diagnostic targets. Some relevant lncRNAs and their involvement in different neurodegenerative diseases have been summarized in Table 1.

## 5. LncRNA as a Potential Target for the Diagnosis and Treatment of Neurodegenerative Diseases

### 5.1. Potential Biomarkers

Aberrant lncRNA expression has been used as a diagnostic tool and extensively researched in metabolic diseases such as cancer. Detection of lncRNA markers has emerged as a critical area in neurodegenerative diseases (NDD) research, which helps in offering insight on the intricate molecular mechanisms underlying conditions such as Alzheimer’s disease, Parkinson’s disease, and ALS. RT-PCR and Z-probe (RNAScope) are the methods used to detect lncRNA in a tissue [122]. Leveraging advanced sequencing technologies, researchers have identified specific lncRNAs associated with neurodegeneration, such as BACE1-AS in Alzheimer’s and NEAT1 in Parkinson’s disease. Bioinformatics tools including machine learning algorithms facilitate the analysis of extensive data sets, adding in the identification of potential lncRNA biomarkers linked to disease progression. CRISPR-based techniques also enhance our understanding of the functional roles of lncRNAs in NDDs [85,123,124].

LncRNA biomarkers can be found in various specimens such as blood, plasma, or CSF. Blood and plasma, due to their non-invasive collection methods, are frequently utilized for biomarker discovery and have been instrumental in identifying lncRNAs linkage to diseases. LncRNA biomarkers in CSF have been of interest, particularly in the context of neurological diseases affecting the central nervous system. For example, studies examining lncRNA expression in CSF have contributed to the understanding of Alzheimer’s disease and Parkinson’s disease [102,103,125].

Chen et al., cited different studies showing increased lncRNA NEAT1 to activate apoptosis and autophagy in diabetic rats with myocardial ischemia-reperfusion injury, which makes NEAT1 a potential biomarker to control cell death processes [101]. There is also the case of the lncRNA GAS5, which is a well-known tumor-suppressive lncRNA that suppresses glioalignancy via an miR-196a-5p/FOXO1feedback loop and has a negative control through miR-18a-5p over migration, invasion, and proliferation [126]. Huaying and collaborators verified, through a quantitative real time-polymerase chain reaction, that lncRNA PART1 is downregulated and lncRNA SNHG14 is upregulated in AD serum samples and can be differentiated from normal samples and is, therefore, used as a biomarker. This study shows five lncRNAs, SNHG14, PART1, NNT-AS1, AC093010.3, and ARMCX5-GPRASP2, that have been found to be differentially expressed in AD and were revealed as potential biomarkers [43]. Yilmaz and collaborators demonstrated that miR-106b-5p was associated with AD susceptibility and is another potential blood biomarker for AD [43,127]. lncRNA CTA-929C8 is highly expressed in the brain compared to normal tissue and was found to be a potential biomarker for AD [55]. Many diseases arising from advanced age are characterized by aberrant transcripts, which could be either coding or noncoding [23]. NEAT1 and MALAT1 are both differentially expressed in ALS, Parkinson’s disease, and Alzheimer’s disease and could be used as biomarkers for these neurodegenerative diseases. With the increase in the aging population, it is necessary to find an effective and rapid diagnostic tool, and lncRNA lends itself to this purpose due to its expression in human bodily fluids.

### 5.2. Treatment

LncRNA that provides adverse effects or promotes the progression of AD through high expression can be silenced by using siRNA specific to target lncRNA. This is observed when siRNA is used to silence BACE1-AS, which then attenuates the effects of BACE1 and improves symptoms affecting memory and learning behavior in animal models [55]. BC200 is expressed at high levels in AD, and the blocking of BC200 by siRNA is assumed to be an effective treatment [52]. Shi Y et al., demonstrated several genes and lncRNAs that are aberrantly expressed in AD, which can lead to an understanding of the mechanisms of disease progression [55]. A study by Wang and collaborators in 2019 revealed that the knockdown of SNHG14 exhibits neuronal protective effects by repressing KRENEN1 and acting as a ceRNA for miR-137 [43]. Knockdown of NEAT1 has been proven to reduce apoptosis and p-Tau levels caused by Aβ and acts as a sponge for miR-107, whose role is to reduce Aβ-induced injuries, thus showing NEAT1 as regulating AD pathogenesis through multiple mechanisms [35].

## 6. Conclusions and Future Perspectives

In neurodegenerative diseases, genetic factors, in addition to aging and environ mental factors, play an important role in the onset and progression of disease. Genetically, Alzheimer’s disease (AD) is divided into two categories: sporadic AD (SAD) and familial AD (FAD). The majority of AD cases are sporadic, but it is affected by several genes. Of the genetic variants that are known to cause the AD include presenilin 1 (PSEN1), presenilin 2 (PSEN2), and amyloid precursor protein (APP) [128]. The Apolipoprotein E (APOE) ε4 allele is reported to be the strongest genetic risk factor for SAD [129]. Additionally, other contributing loci that affect AD include triggering receptors expressed on myeloid cells 2 (TREM2), sortilin-related receptor 1 (SORL1), and adenosine triphosphate-binding cassette transporter subfamily A member 7 (ABCA7) [128,130,131]. Similar to AD, genetic factors also increase the risk for the development of Parkinson’s disease (PD). Several genes including SNCA, LRRK2, VPS35, PRKN, PINK1, GBA, and DJ-1 have been convincingly associated with PD [132,133]. In line with AD and PD, most amyotrophic lateral sclerosis (ALS) cases are sporadic, and only 5–10% of ALS cases were associated with genetic factors; although, more than 40 genes have been identified that have been linked with ALS. However, mutations in four genes C9ORF72, SOD1, TARDBP, and FUS account for the majority of familial ALS cases [134]. The noncoding part of the genome, which comprises more than 98% of the genome, has been getting more attention in terms of the causes, diagnosis, and therapeutic aspects of NDD.

Knowledge on LncRNAs has changed dramatically over the past decade, having developed rapidly and achieved significant successes, and is likely to continue to expand in the future. There has been a noticeable surge in the attention given to lncRNAs in the field of biomedical research, particularly concerning NDDs. This surge in interest is reflected in the substantial increase in the number of scientific articles focused on both lncRNAs and NDDs. This indicates a growing recognition of the importance of these non-coding RNA molecules in the context of neurological health and dysfunction. With the increase in lncRNA shown to be dysregulated in AD, there may be novel pathways to target.

There are various functions of lncRNAs, which include translation, post-translation, and epigenetic modifications. There are a few issues or challenges that need to be overcome in lncRNA-targeted therapy for neurodegenerative diseases. The major challenge is identifying the lncRNA with the most significant potential, which is also the basis for the final clinical application of targeted lncRNA drugs. In addition, it is necessary to develop highly specific targeting methods and precise delivery methods, i.e., how to deliver the therapeutic agent to the target tissue through the blood–brain barrier and/or how to overcome the degradation of enzymes, i.e., nuclease and/or ineffective endocytosis of target cells and/or their clearance from the body. Despite all the challenges of exploring the role of lncRNAs in neurodegenerative diseases, the future is very significant.

## Figures and Tables

**Figure 2 ijms-25-02268-f002:**
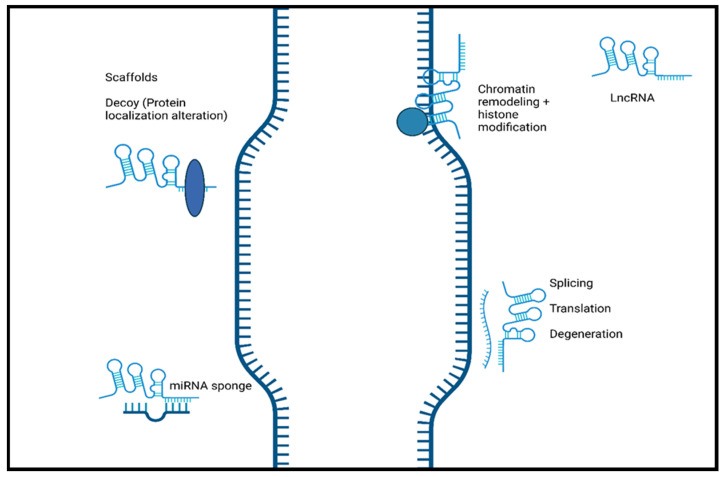
The schematic representation of the function of long non-coding RNA. LncRNAs induce chromatin remodeling and histone modification. Interaction with mRNA. LncRNA hybridization may lead to alternatively spliced transcripts, translation, and mRNA degeneration. LncRNAs interact with proteins/biological molecules to modulate their activity by binding to specific proteins and altering protein localization. They also serve as scaffolds to allow the formation of miRNA sponges. Created with BioRender.com.

**Figure 3 ijms-25-02268-f003:**
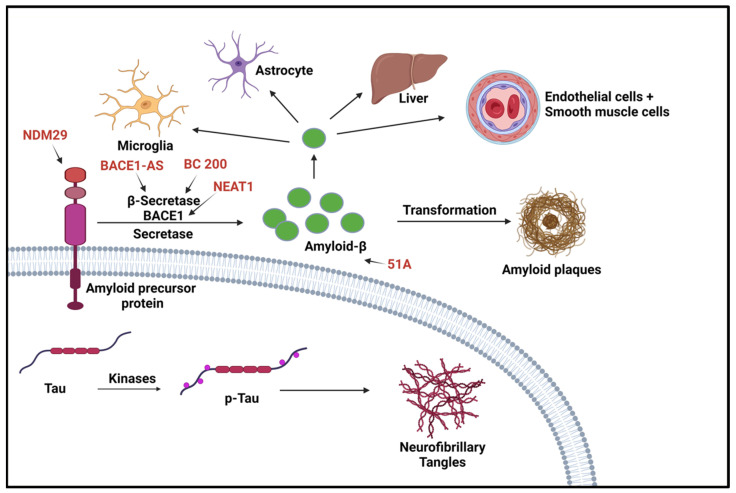
LncRNAs involved in Alzheimer’s disease. Two aspects of this disease are shown here. The first aspect is the production/aggregation of Amyloid-β (Aβ) peptides, which are the final product of amyloid precursor proteins (APPs); the other aspect is the accumulation of neurofibrillary tangles (NFTs) caused by hyperphosphorylated microtubule-associated protein (p-Tau). BACE1 is the rate-limiting enzyme for amyloid precursor proteolysis and is affected by BC200 and BAEC1-AS. NDM29 induces the formation of APP, which in turn promotes an increase in Aβ formation [11]. “Created with BioRender.com”.

**Figure 4 ijms-25-02268-f004:**
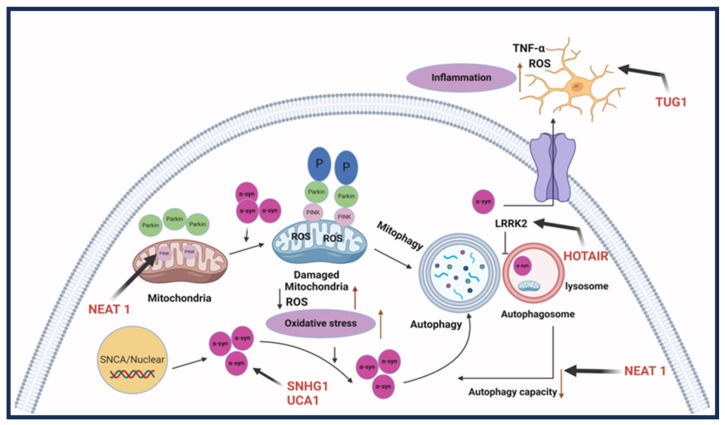
LncRNAs involved in Parkinson’s disease. Parkinson’s disease pathogenesis involves three aspects: the first is dopaminergic neuron death; the second is an aggregation of synuclein-alpha, which forms Lewy bodies; and the final one is neuroinflammation, causing cell death. LncRNA HOTAIR, SNHG1, and UCHL1-AS participate in the accumulation of SNCA. LncRNA NEAT1 causes abnormal autophagy. HOTAIR can also upregulate LRRK2 and thus affect autophagy. LncRNA TUG1 regulates microglia polarization and increases inflammatory cytokine production [11]. Created with BioRender.com.

**Figure 5 ijms-25-02268-f005:**
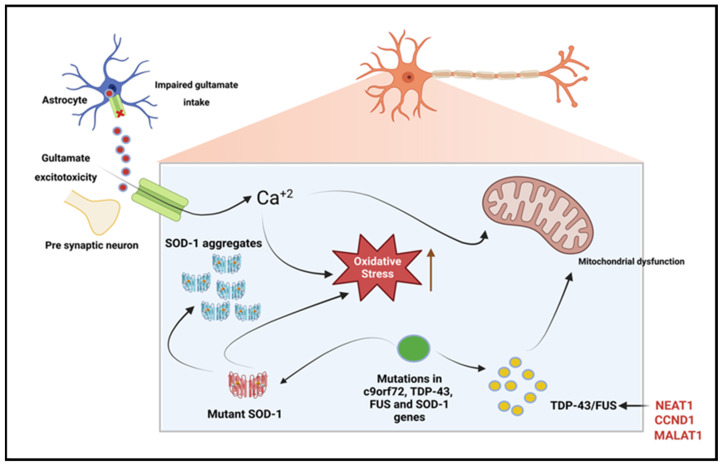
LncRNAs involved in amyotrophic lateral sclerosis. Due to dysfunction of the astrocytic excitatory amino acid transporter 2 (EAAT2), there is reduced uptake of glutamate from the synaptic cleft, which leads to glutamate excitotoxicity. The resulting glutamate-induced excitotoxicity induces neurodegeneration through the activation of Ca^+2^ dependent pathways. TDP-43, c9orf72, and fused in sarcoma (FUS) gene mutations result in dysregulated RNA metabolism, leading to the formation of intracellular neuronal aggregates. Superoxide dismutase-1 (SOD-1) gene mutations increase oxidative stress and induce mitochondrial dysfunction, leading to intracellular aggregates. Enriched lncRNA NEAT1 is proved to be the target of both FUS and TDP-43 [97]. Created with BioRender.com.

**Table 1 ijms-25-02268-t001:** Summary of relevant lncRNAs and their involvement in different neurodegenerative diseases.

LncRNA	Type of NDD	Notes	References
BACE1-AS	Alzheimer’s	BACE1-AS has been associated with regulation of BACE1, a key enzyme in amyloid β production	[53,102,103]
NEAT1	Parkinson’s	Associated with modulation of neuronal apoptosis and neuro-inflammation	[53,104,105]
MALAT1	Alzheimer’s	Linked with neuronal apoptosis and neuro-inflammation	[48,106,107]
SNHG1	Alzheimer’s/Parkinson’s	Plays a role in regulation of amyloid β production and neuro-inflammation	[108,109,110]
ANRIL	Parkinson’s	Associated with vascular dysfunction and inflammation in CNS	[111]
HOTAIR	Alzheimer’s	Associated with dysregulation of synaptic plasticity and neuronal apoptosis, contributing to cognitive decline	[112,113]
TUG1	Parkinson’s	Shown to modulate dopaminergic neuronal cell death suggesting its involvement in pathogenesis of Parkinson’s	[70,114]
BC200	Parkinson’s/Multiple sclerosis	Involved in regulating mRNA translation and synaptic plasticity and contributing to disease progression	[115,116]
MEG3	Alzheimer’s	Implicated in amyloid β-induced neurotoxicity and neuronal apoptosis	[117]
PINK-AS	Parkinson’s	Impairment of mitochondrial dynamics due to decrease in the PINK1-AS and neurodegeneration due to ASUCHL1 downregulation	[118]
NDM29	Alzheimer’s	NDM29 expression is enhanced in the cerebral cortex of AD patients	[45,119]
LRP1-AS	Alzheimer’s	LRP1 is deeply involved in APP trafficking and Aβ processing	[120,121]

## Data Availability

Not applicable.

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
