# Peer review of "Long Non-Coding RNAs: New Insights in Neurodegenerative Diseases"

_ijms, 2024, doi:10.3390/ijms25042268_

Round 1
Reviewer 1 Report
Comments and Suggestions for Authors
My suggestions:
1. I would add a short chapter about the detection of lncRNA biomarkers.
2. I would add a table on lncRNAs, which may involved in different neurodegenerative diseases.
3. I would mention some genetic factors for each disease.
4. It would be nice to discuss, from which biospecimens the lncRNA biomarkers could be detected (blood, plasma, CSF, etc)
5. Were LncRNAs studied in multiple sclerosis, small vessel diseases, or leukoencephalopathies?
6. Do lncRNA therapies have any side effects?
Author Response
Resubmission ijms-2831576
Reviewer #1
We thank the reviewer for the positive feedback, constructive criticism, and suggestions. We have incorporated all the suggestions. Response in summary is mentioned here below:
Comment #1. I would add a short chapter about the detection of lncRNA biomarkers.
Answer: We appreciate the reviewer's suggestion to make the manuscript better. We have added “detection of lncRNA biomarkers” accordingly in the revised submission under section 5.1, with references.
Comment # 2. I would add a table on lncRNAs, which may involved in different neurodegenerative diseases.
Answer: Thanks for the observation and suggestion. We have added Table 1 with the appropriate references.
Comment # 3. I would mention some genetic factors for each disease.
Answer: We sincerely thank the reviewer for the insightful suggestion; we have added the genetic factors of each disease in section 6, with references.
Comment # 4. It would be nice to discuss from which biospecimens the lncRNA biomarkers could be detected (blood, plasma, CSF, etc.)
Answer: We sincerely thank the reviewer for the insightful suggestion; we have added the biospecimens from which the lncRNA could be detected in section 5, with references.
Comment # 5. Were LncRNAs studied in multiple sclerosis, small vessel diseases, or leukoencephalopathies?
Answer: Thanks for the relevant question.
LncRNAs regulate several biological functions and play a critical role in health and disease. Several studies have demonstrated the role of lncRNAs in multiple sclerosis (PMID: 34659340; 30497529) and small vessel diseases (PMID: 33340430; 29896231). The purpose of this review article is to summarize the current knowledge on the role of lncRNA in most common neurodegenerative diseases that share similar clinical features (proteinopathies). In the future, we will extend our work on describing the role of LncRNAs in demyelinating and small vessel diseases.
Comment # 6. Do lncRNA therapies have any side effects?
Answer: We thank the reviewer for the valid question.
Several RNA-based therapies have been approved for clinical use in recent years, with many currently under consideration (PMID: 36841158; 34145432). The natural ability of lncRNAs to regulate diverse biological functions underpins their potential as an effective candidate for therapy. Several preclinical studies provided proof-of-concept evidence that lncRNA-based therapies can be effective in preventing the progression of diseases with minimal to no adverse effects (PMID: 33427561; 27333023; 34083547). However, more rigorous studies are warranted to examine the safety profile, brain permeability, and off-target effect before translating these successes into humans.
Reviewer 2 Report
Comments and Suggestions for Authors
In the article entitled "Long Non-coding RNAs: New Insights in Neurodegenerative Disesases", the authors review long non-coding RNAs and their implications in degenerative diseases such as Alzheimer's, Parkinson’s, and amyotrophic lateral sclerosis.
The topic is interesting and actual, however, there are several articles that are similar to the review.
It would be important for the authors to point out the novelty of the review.
It is recommended to make a table with non-coding RNAs and the disease.
It is recommended to change the first sentence of the introduction, line 36, page 1.
Figure 3 seems to have a spelling error, "Aamyloid-B", is this correct?
Author Response
Resubmission ijms-2831576
Reviewer # 2
In the article "Long Non-coding RNAs: New Insights in Neurodegenerative Disease," the authors review long non-coding RNAs and their implications in degenerative diseases such as Alzheimer's, Parkinson’s, and amyotrophic lateral sclerosis. The topic is exciting and actual. However, there are several articles that are similar to the review.
We sincerely appreciate the reviewer’s time to help make the article better. We have revised the article as per the reviewer’s suggestions. The revised manuscript has the changes in blue color. Please find the point-wise answers down below:
Comment# 1. It would be important for the authors to point out the novelty of the review.
Answer: Sincerely,
The novelty of this review article lies in summarizing the recent research progress on lncRNAs playing a functional role in the pathogenesis of neurodegenerative diseases. We further discuss the associations of lncRNAs with Alzheimer’s disease, Parkinson’s disease, and amyotrophic lateral sclerosis through several mechanisms, such as neuroinflammation, amyloid-beta deposition, toxic tau, and α-synuclein accumulation. We also discuss potential targets of lncRNAs and their promise as novel therapeutics. Also, the article leads towards novel genetic biomarkers that can be used as exclusionary tools specific to NDDs. We hope that our review will be helpful in addressing this question and become a resource for future studies based on lncRNAs in neurodegeneration.
Comment # 2. It is recommended to make a table with non-coding RNAs and the disease.
Answer: Thanks for the recommendation. We have added the table (Table 1), with references.
Comment # 3. It is recommended to change the first sentence of the introduction, line 36, page 1.
Answer: Thanks for the suggestion. We have changed the first sentence as suggested by the reviewer.
Comment # 4. Figure 3 seems to have a spelling error, "Aamyloid-B", is this correct?
Answer: Thanks for looking at the figure carefully. We have corrected it and have it in the resubmission.

Round 2
Reviewer 1 Report
Comments and Suggestions for Authors
The authors fulfilled my suggestion
Reviewer 2 Report
Comments and Suggestions for Authors
Thanks to the authors for the answers.
The authors did substantial changes to the manuscript so it improved.